# How can school help victims of violence? Evaluation of online training for European schools' staff from a multidisciplinary approach

**Ana M. Greco**[1,2]*, **Carla González-Pío**[1], **Marina Bartolomé**[1,3], **Noemí Pereda**[1,4], **on behalf of SAVE Project Team**¶

**1** Grup de Recerca en Victimització Infantil i Adolescent (GReViA), Universitat de Barcelona, Barcelona, Spain, **2** Estudis de Dret i Ciència Política, Universitat Oberta de Catalunya, Barcelona, Spain, **3** Departament dse Formació, Fundació Vicki Bernadet, Barcelona, Spain, **4** Departament de Psicologia Clínca i Psicobiologia, Universitat de Barcelona, Barcelona, Spain

¶ Membership of the SAVE Project is provided in the Acknowledgments
* agreco0@uoc.edu

**Data Availability Statement:** All relevant files (i.e., database, code, instrument protocols) are available from the Open Science Framework database in the

## Abstract

The interventions used to prevent or treat violence against children, particularly sexual abuse, tend to only consider the target audience as their main source of data. We tested the effect of an online training for school staff members in Europe through three studies. In Study I, we interviewed 5 adult women ($M_{age}$ = 49.2, $SD$ = 5.81) who were victims of sexual abuse during childhood to assess what school could have done during that time to protect them. Through Study II, we collected data on 66 school staff members to assess feasibility (based on quantitative indicators) and to explore the changes they would make to their everyday practice due to the training course (using qualitative analysis). In Study III, we used network analysis to assess to what extent the actions described by school staff in Study II met the needs expressed by the victims in Study I. Findings of Study I revealed new proposals from the victims' perspective, such as working with the perpetrators. Study II showed the feasibility of training and identified five types of action that school staff members will include in their everyday working dynamics due to the training: detection (e.g., *Greater attention to relationships with peers*), reporting (e.g., *Now I know that suspecting a case of child abuse is enough to report*), everyday practices (e.g., *Introducing a calming space*), changes at school level (e.g., *Propose the training course to the school management team*) or practices that could belong to more than one category (e.g., *Greater awareness of the activities undertaken by the school*). Study III provided evidence that some of these changes (e.g., reporting without looking for proof) were in line with some of the victims' expectations (e.g., listen to the children). We also identified gaps that need to be further developed.

following website: https://osf.io/7q3rt/ Due to the sensibility of data, personal information has been removed.

**Funding:** This study was funded by the European Commission through their Erasmus+ program, strategic partnership K201, 2018 call (Grant number: 2018-1-ES01-KA201-050287). The funders had no role in study design, data collection and analysis, decision to publish, or preparation of the manuscript. Author Carla González-Pío received a salary from the funder.

**Competing interests:** The authors have declared that no competing interests exist.

## Introduction

In 1996, the World Health Organization declared violence a major human rights, public health, and social problem [1]. In this line, interpersonal violence, or victimization, has been defined as "harm that occurs to individuals due to other human actors behaving in ways that violate social norms" [2]. The malevolence of the act caused by another human being, the feeling of betrayal and injustice, and the transgression of established social norms, confers a special traumatogenic potential to experiences of interpersonal violence [3]. The developmental victimology theory defends that children and adolescents are vulnerable to the same forms of victimization as adults (like robbery or assault) and other forms that are specific to their age group related to the dependency from the adult (such as, physical abuse or neglect by their caregivers) [2]. Thus, multiple forms of victimization seem to affect children and adolescents which can be grouped as peer victimization, caregiver victimization, conventional crimes, sexual victimization, electronic victimization, and exposure to violence or indirect victimization. From this perspective, many studies proved the high prevalence of various types of violence against children globally [4–8], and their devastating effects on victims across their lifespan [9,10] and on society as a whole[11–13]. In Europe, the median rates of types of violence against children range from 6.2% (sexual abuse of boys) to 27.0% (emotional abuse of girls) [14].

Multiple studies suggest that suffering victimization in these evolutionary stages is related to difficulties in interpersonal relationships [15] and a poorer level of well-being and psychological health, such as behavioral or emotional problems [16]. It also appears to have a negative impact on the academic environment. In a meta-analysis of 30 studies on the behavior exhibited by children experiencing neglect or emotional abuse, 12 showed cognition and academic differences in performance among children who had been neglected and/or emotionally abused in comparison to their counterparts who were not exposed to violence [17]. In addition, a relationship was found with low general intelligence and a worse performance in literacy and arithmetic skills. In a study with 5,930 participants between 12–18 years of age, those who had experienced victimization on a regular basis were at greater risk of having lower academic performance, which was partially mediated by absenteeism [18]. Systematic reviews and meta-analyses of 67 and 43 studies respectively showed that all forms of violence in childhood have a significant impact on educational outcomes [12]. According to this research, victims seem to be less likely to graduate from high school than students who never experience violence. Boys who are bullied are nearly three times more likely to be absent from school than those who are not. Girls who have experienced sexual violence have a three-fold increased risk of being absent and present lower academic achievement on standardized tests than girls who have never been through this kind of experience [12]. Although all these effects could also be warning signs for school staff, prevention, detection and intervention when children may be at risk of violence is still very challenging to act against violence within this environment [13,19,20]. Despite a considerable proportion of school staff detecting potential cases of violence [21,22], most of these concerns are never reported to external agencies [20,23]. As a result, victims do not receive specialized help and support [19,24] or do not disclose their experiences [25]. Many reasons for this behavior have been described by previous studies, such as low levels of knowledge or poor training, fear of the consequences for the victim, the family, the reporter and the school or lack of support from the school management team [26–29].

The relevance of prevention has been outlined, since everyday school actions other than reporting seem to make victims feel safe at school and encourage them to talk about their issues [30]. Caring about students' well-being and mental health, creating a school environment that is safe and trustworthy or taking time with each student to discuss emotions or

provide strategies to foster resilience and post-traumatic growth are a few strategies that seem to work effectively and are usually embraced by trauma-sensitive schools [31–33]. Interventions that have been tested in this approach have promising effects in terms of improvements in teachers' knowledge about child abuse and neglect [34], students' school adjustment and mental health outcomes [35] and in the reduction of violent experiences [36]. Some studies have also found that training teachers in the prevention of child victimization enables students to progress in learning [37] and benefits the early learning skills of children and adolescents, improving the quality of the school environment [38].

Finally, some studies have shown the importance of considering the voice of the victims [36,39,40] to take into account their needs, as they must be tailored to suit their individual situations [19,24] in the violence prevention programs and the promotion of resilience practices at schools (40). This seems to be particularly true for victims of sexual abuse. It has been shown that sexually abused children who feel safe and communicate effectively with their parents and their protection network tend to believe that adults and the world around them are trustworthy, and therefore may be more prone to disclose their experience of abuse [41]. However, there are many reasons not to disclose this experience, including personal factors related to the victim and the established dynamic with the abuser and, most importantly, factors related to the social reaction to the experience of sexual abuse [42,43]. Various studies confirm that this type of violence seems to elicit negative reactions among people who receive the disclosures, such as blaming or disbelieving the victim [44–46]. This tends to have deep negative effects on the mental health of the victims who dare to disclose the abuse (44) and can lead to a harder healing process in adulthood [47]. Fear to these reactions is probably the reason why most victims tend to keep silent during childhood [48,49], and what makes sexual abuse still one of the least officially reported types of violence against children in Europe [50,51]. A critical review on social reactions to child sexual abuse disclosures [52] concludes that a large group of victims will not be able to explain the abuse to anyone in their lifetime.

Programs about child abuse addressed to teachers and studies performed with school staff [34,53,54] report that this population lack training to prevent, detect and intervene effectively when faced with victimization against children and adolescents. On the other hand, studies testing programs [55] including the victim's voice [39,40] or the trauma-sensitive approach [36,56] have found positive effects at the school level (e.g., the reduction of violent incidents or improvements in the school climate) and the individual level (e.g., decreasing post-traumatic stress symptoms or increasing hope and school adjustment). Based on this evidence, the current research presents and tests a new online intervention: Schools Against Victimization from an Early Age (SAVE) through an innovative methodology. SAVE's main objective is to develop education staff's skills to support and protect children who have experienced some form of violence in Europe. As it was conceived to address the entire European region, SAVE was created by five expert partners institutions from countries representing different European backgrounds (i.e., Eastern, Mediterranean, and Western Europe). SAVE is an online course that includes information about prevalence, signs, reporting procedures and other actions to prevent, detect and report potential cases of child sexual abuse, maltreatment and bullying. It also proposes a module about resilience, which has the aim to help educators to implement tools in the classroom so their students can grow from their traumatic experiences, such as recognizing their strengths and help them to manage their emotions. The course also includes additional material to use in school; protocols to deal with disclosure or suspicion of abuse; stories, videos, songs and activities that could be shared with the children; a list of NGOs dealing with abuse in each participating country, information about requirements of the legal framework and presentations about SAVE project's development and results. SAVE has been funded by

the European Union through its Erasmus+ program, is entirely free, open access and available in five European languages at https://saveprogramme.eu/.

Given the lack of programs with SAVE's characteristics and the previously reviewed evidence, the aims of the present study were:

1. To explore how victims of sexual abuse during childhood remember the school's role in their lives at the time, to try to find out what helped them and what made it harder for them.

2. To assess the potential of the SAVE course through the feedback given by school staff members about the whole course and through victims' of child sexual abuse feedback about the sexual victimization module. This objective specifically aims

   a. To assess schools' potential in the prevention, detection and reporting of child victimization through changes that the staff of educational centers would include in their everyday practice as a result of the course.

   b. To test to what extent the actions proposed by school staff members meet the needs described by victims regarding the school's role when they suffered child sexual abuse.

In line with these research aims, we performed three studies. Study I addressed the first research question by interviewing victims of child sexual abuse about their school experience. Study II addressed research goal 2.a. by analyzing members of European schools' staff's impressions of the course, particularly the strengths and weaknesses and whether they would make some changes in their working life as a result of taking the course. To combine the findings of both studies, we compared them in Study III, in which we tested whether what was described by school staff in Study II could respond to what victims of sexual abuse claimed could have been helpful during their school experience in Study I. This third study aimed to meet research aim 2.b.

## Materials and methods

### SAVE design and testing

As previously mentioned, SAVE was created by a partnership of five European entities based in four different European countries: Ireland (ICEP Europe), Italy (Università degli Studi di Padova), Poland (Fundacja Dajemy Dzieciom Siłę) and Spain (Fundació Vicki Bernadet and Universitat de Barcelona), each of which is specialized in one of the aspects that the course addresses. Each partner was in charge of creating a module on their expertise, as shown in Fig 1. To maintain consistency across the course, the content and structure of each SAVE module was agreed by all members of the partnership.

A challenging aspect of SAVE was that it had to be applicable across Europe but also include previous findings about this issue in different localities [57–59], so we had to take into account the particularities of each region, such as the official languages, the different procedures to report child victimization or the required training to become a teacher or any other type of school staff members in each country. To overcome this challenge, all members of the partnership discussed applicability in their region and reviewed the modules drafted by the other partners to check for coherence and a common basis. In addition, each partner included a section about local resources to consider in each country.

Another innovative aspect of SAVE is that it combines stakeholders' perspectives, since it is composed of universities, non-governmental organizations (NGOs) and a small company. SAVE is strong in its diversity beyond the partnership, as the program was tested through a

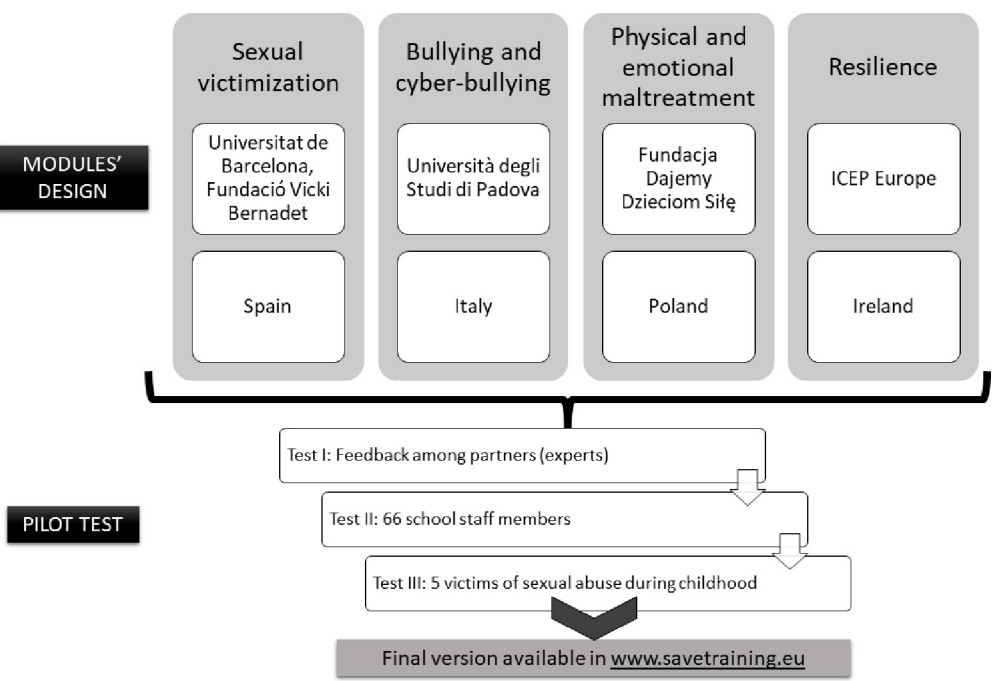

**Fig 1. Design and testing of SAVE course.** Each module was composed of an introduction page, a real story, definition, prevalence, signs to detect potential cases, what to do, reporting procedures, prevention, self-assessment, summary and references.

pilot study by school staff members and adults who were victims of sexual abuse during childhood and was reviewed by external experts (AUGEO Academy, www.augeo.nl/en). With feedback from these agencies, SAVE could not only be validated but also enriched and improved. Fig 1 summarizes the design and testing of SAVE training. The course and all additional materials are fully and freely accessible in www.savetraining.eu.

## Sample

**Study I.**   Five women who were victims of sexual abuse during childhood between 42 and 55 years old ($M_{age}$ = 49.2, $SD$ = 5.81) participated in Study I. All of them were Spanish and had achieved a level of studies beyond compulsory high school (four of them had a university degree and one a tertiary diploma). Two of them were unemployed, one was on leave, one employed in a non-qualified position and the other in a qualified one. All of them were mothers of at least one child. One of them was married, two of them had a current partner (both different from their child's father) and two of them were single.

Regarding the experience of abuse, the age of onset ranged from 3 to 9 years old ($M$ = 6.75, $SD$ = 2.87), although one participant declared that she did not recall when it had started. All of them had been abused by members of their families and in two cases they had had more than one abuser. The difference in age with the first abuser ($n$ = 3), when reported, ranged from 7 to 16 years ($M$ = 11, $SD$ = 4.59). More details about participants of this study are shown in Table 1.

**Study II.**   Sixty-six school workers engaged in study II. Most were highly educated women. Their age ranged from 21 to 64. All participants lived in Europe; over ninety percent of the subjects ($n$ = 64, 93.93%) had European nationalities from countries participating in the project; one was from Venezuela and another from the Philippines. Most of them (77.27%)

**Table 1. Participants who were victims of CSA (Study I and Study II).**

| Participant | 1 | 2 | 3 | 4 | 5 |
|---|---|---|---|---|---|
| *Sociodemographic variables* | | | | | |
| Age | 52 | 44 | 55 | 53 | 42 |
| Nationality | Spanish | Spanish | Spanish | Spanish | Spanish |
| Education | University | University | Tertiary | University | University |
| Occupation | Qualified worker | On sick leave | Unemployed | Employee | Unemployed |
| *Variables related with the experience of abuse* | | | | | |
| Age of onset | Around 9 years old | Around 9 years old | Unsure but from a very early age | 6 years old | 3 years old |
| Abuser | Her brother | Her cousin, who was also her brother-in-law | The first one person was her brother but also her brothers' friends, brother-in-law, a cinema employee and teachers. | Her father | Her grandfather and a peer at school |
| Family | She left home with her mother when she was 18 years old. | Her sister found out but did not believe her. | Everybody in her family was aware. She once told her mother, who punished her for lying. | Her sister was also abused by her father. They were born in Brazil and came to Spain only with her mother. | No data |

had a university degree. In terms of their professional profiles, over 59% of participants were teacher, instructor, or professors, and most of them (68.18%) were working in the public sector. Other details are shown in Table 2.

**Study III.** This study aimed to merge the findings of studies I and II, so we included all participants from both previous studies.

## Procedure

All data collection procedures were in accordance with ethical recommendations and the study was approved by the university's bioethics committees (IRB 00003099).

For Study I, potential participants who were in treatment for their sexual abuse experience in an NGO were contacted by author MB, who works in the institution. Inclusion criteria was having been abused continuously before the age of 18 years old by a family member or a trusted adult. It was also considered whether the therapist of the NGO thought that participation in the study was suitable for their patient. Potential participants were told about the aims and characteristics of the study and were asked if they wanted to take part. Five of the eleven people who were invited agreed to participate (45.45% response rate). At this stage, participants were sent an e-mail including a document to remind them of the aims of the research, explaining in more detail the conditions of their participation, the use of the data they would provide, and their data protection rights. In this same e-mail the link to the module of sexual victimization of SAVE course was included, so participants could freely navigate through it at least a week before the interview. It was highlighted that their collaboration was voluntary and that it would not have any effect on the treatment and services they were receiving at the institution. It was also specified that data would remain confidential, and their identities would only be accessible to the member of the research team who was responsible for their participation. A consent sheet was included in the document for them to sign if they agreed to participate under the agreed terms. Participants also had the option to give oral consent right before the interview. A virtual interview was then set up based on their availability and preferences but always at least a week after receiving the invitation e-mail. Participants were asked to send the signed document or provide verbal consent before the interview took place. All the participants gave their informed consent by signing the document and sending it by e-mail before the interview was held. The interviews would be held without a camera and participants were given the choice of not being recorded, if they preferred. The research team engaged to destroy

**Table 2. School staff participants (Study II).**

|  | *n* | % |
|---|---|---|
| *Gender* | | |
| Men | 8 | 12.12 |
| Women | 58 | 87.87 |
| *Age*[a] | | |
| 21–34 | 22 | 33.33 |
| 35–44 | 24 | 36.36 |
| 45–54 | 14 | 21.21 |
| 55 or more | 6 | 9.09 |
| *Nationality* | | |
| Italian | 5 | 7.57 |
| Irish | 8 | 12.12 |
| Polish | 17 | 25.75 |
| Spanish | 34 | 51.51 |
| Other non-European nationalities | 2 | 3.03 |
| *Level of education* | | |
| High school | 3 | 4.54 |
| Tertiary | 3 | 4.54 |
| University or beyond (Master, PhD) | 51 | 77.27 |
| Others (Sports or Ballet studies, Professional training) | 9 | 13.63 |
| *Occupation* | | |
| Teacher, instructor or professor | 39 | 59.09 |
| Psychologist, educator, special needs assistant | 19 | 28.78 |
| Others[b] | 8 | 12.12 |
| *Type of institution in which their work* | | |
| Public [c] | 45 | 68.18 |
| Semi-private | 10 | 15.15 |
| Private | 6 | 9.09 |
| Non-profit organization | 5 | 7.57 |
| *Age of children their work with*[d] | | |
| Less than 3 years old | 2 | 3.03 |
| 3 to 5 years old | 17 | 25.75 |
| 6 to 12 years old | 46 | 69.69 |
| 3 to 12 years old | 48 | 72.72 |
| Over 12 years old | 28 | 42.42 |

[a] Age was categorized according to Levinson [60].

[b] In this category were included school staff that were hired to do replacements and guardians.

[c] This category included workers hired by private companies that served in public institutions.

[d] Percentages do not add up to 100% because respondents were able to select more than one category.

the records of their interviews once the research had been completed. Participants were offered a summary of the results and were invited to participate in the analysis (see the Methods section).

For Study II, the general data collection procedure was agreed by all representatives from each of the partner institutions of SAVE. The project manager in each country contacted potential participants in their region and informed them about the project and the study that was being undertaken. A total of 38 participants were excluded, according to the following

exclusion criteria: absence of consent (5.26%), inactive work status (2.63%), they did not respond to certain items (86.84%) and duplicate participants (5.26%). All partner institutions posted the study on their social media and potential participants contacted the institution to collaborate. As another way to invite potential participants, people who had already participated in a similar project and agreed to be contacted to participate in similar initiatives were e-mailed and asked to collaborate. An e-mail was sent to participants who expressed their willingness to collaborate, informing them of the aims and conditions of their collaboration and with a link to a questionnaire on the Qualtrics platform (www.qualtrcis.com). The first page of the questionnaire described the aims of the study, the characteristics of the collaboration, stated that data would remain confidential and noted that participants were entitled to withdraw at any time with no consequences.

For participants of both studies, contact details of the relevant member of the research team were included in case they wanted to ask questions or make comments or suggestions.

## Instrument

Study I had an exploratory aim and a design that involved in-depth interviews. A semi-structured interview script was developed by members of the research team. Scripts mostly included open-ended questions on the experience of abuse, memories of school while the abuse was occurring, whether they could remember anything that particularly helped or stressed them about their school experience during that time, and how they thought school could have helped them. They were also asked about several characteristics of the course content, specifically on the sexual victimization module (i.e., whether they found it useful, understandable, applicable, innovative, realistic, suitable, relevant and respectful) and they were invited throughout the interview to make comments or suggestions.

For Study II, data were gathered through an ad hoc self-report questionnaire that included: a) a multiple-choice item about course length (i.e., if it was too short, the right length or too long), and two subscales of four items each in which participants had to rate their level of agreement on a 5-point Likert scale (ranging from 1 to 5) with statements about the quality of the environment (e.g., "it is user friendly") and content (e.g., "it is understandable"). Cronbach's alpha for each of these subscales was .93 and .96 respectively. Participants were also asked through a brief open-ended item to describe a change that they were going to include in their everyday work as a result of this training course and were invited to add any further comments or suggestions. The questionnaire was applied through Qualtrics (www.qualtrcis.com) platform, as provided by the University of Barcelona.

In both studies, a sheet was included to collect sociodemographic and professional data. The full instruments protocols are available in https://osf.io/7q3rt/

For Study III, the information gathered through the previously described instruments were combined.

## Data analysis

**Qualitative analysis.** *Study I.* All recorded interviews were transcribed verbatim and data analysis occurred after data collection. The principles of conventional content analysis [61] were used to guide the process for coding, categorizing and synthesizing the interview data. At the beginning of this process two research team members (AMG and CG) independently read and conducted line-by-line coding of the five transcripts for the interviews to identify emerging codes. These individuals with an extra member of the research team (MB) met to compare, review, define and refine the categories. Two authors (CG and MB) then re-coded all the transcripts independently, and a Kappa coefficient was computed to assess whether the method

produced a consistently meaningful description of the issue across the independent raters, to obtain a $\kappa = .4$, $p < .001$. A third coder (AMG) then re-coded and solved discrepancies between the other two and the inter-reliability of the coding system was re-assessed through the Kappa coefficient to reach a value of $\kappa = .54$ ($p < .001$) between the third and second coder and a value of $\kappa = .73$ ($p < .001$) between the third and first coder. This indicates a moderate and solid level of agreement, respectively [62]. Fragments with no agreement that did not to provide new information were either removed ($n = 4$) or solved by consensus ($n = 2$). Categories were then assigned to each of the 81 fragments analyzed, using the most frequent code among the three raters. Interrelated codes were arranged into subcategories and under broader categories representing emerging themes in the data.

To evaluate the validity of our system, we provided a clear definition of each of the emerging concepts as evidence of construct validity. We also compared how categories were distributed across participants (S1 Fig is provided as supplementary material, in https://osf.io/7q3rt/), as evidence of criterion validity. To confirm the credibility of our interpretation of the comments made in the interviews, summaries of interpreted data were circulated to participants who agreed to take part at this stage for their comments. They were given one week to review data summaries and to provide additional comments. Four out of the five participants confirmed that the interpretation was coherent. One participant categorized independently four fragments of her interview and obtained the same categories in three of them.

*Study II*. Answers to the question "what change will you include in your everyday work thanks to this training?" were categorized based on their content independently by AMG and NP, using the same basis and procedure as for study I [61]. A total of 74 fragments were found in the responses given by the 66 participants. Reliability was tested through the same procedure as for study I, to obtain a value of $\kappa = .78$ after the first round of coding. We considered this substantial agreement and revised the fragments in which the two coders disagreed. On closer inspection, the content of these fragments ($n = 16$, 21.62%) included aspects that could be assigned to two categories (e.g., "I'll refine the procedures" could belong to the category of "Report" but also to the category of "Everyday actions"). Hence, we assigned a fifth category to fragments that could be included in more than one category.

The validity of the system was tested through statistical analysis of correlations between the categories and the sociodemographic variables. Contingency tables along with Fischer's exact test results are provided as supplementary material.

*Study III*. To test whether what was described by school staff was in line with what victims of sexual abuse claimed could have been helpful during their school experience, we merged the findings of studies I and II. On the basis of similarity [63], according to the principles of conventional content in qualitative analysis we linked each action described by school staff with a category described by victims of sexual abuse during childhood. Actions described by participants in study II that could be assigned to one or more categories were excluded ($n = 16$). A network analysis (NA) [64] was then conducted to quantify how many actions were assigned to each category, identifying aspects that were covered by school staff trained through SAVE. The NA has been recently proposed as an effective way of mixing quantitative and qualitative indicators [65,66] and provided relevant findings, such as the identification of categories that needed further development.

**Quantitative analysis.** For Study II, descriptive statistics were obtained for items assessing the length of the course, the quality of the environment and the course content. To compare these scores across characteristics of the participants, $\chi^2$, t-test or ANOVA (along with pairwise post-hoc comparisons) were used.

To store, manage, and organize the data, all transcripts were uploaded to the Open Source R Software [67]. In addition to the standard packages, "psych" [68] and "fmsb" [69] was used

to assess psychometric properties, "ggplot2" [70] to plot analyses, "network" [71,72] "igraph" [73], "visNetwork" [74] and "networkD3" [75] to perform and plot network analyses. R Software [67] was used to perform all analyses. The database and code for kappa coefficient and network analysis are available in https://osf.io/7q3rt/. Personal data has been removed in order to protect participants' privacy.

# Results

## Study I

A total of 81 fragments were extracted and categorized. Table 3 shows the system of emerging categories with their definition, an example and the frequencies.

## Study II

Over 80% of participants considered that the length of the course was correct ($n$ = 55). Some of them added comments asking for in-person courses or stating that the length was adequate for an introductory training session, but they would like to go deeper into the topic. The difference between the proportion of choices did not differ significantly by gender ($\chi2(2)$ = 3.48, p = .18) or nationality ($\chi2(10)$ = 11.27, p = .34), type of institution in which participants worked ($\chi2(6)$ = 5.22, p = .52) or main occupation ($\chi2(4)$ = 4.87, p = .30). People up to 44 years old chose the option "correct" (n = 36) more frequently than people beyond this age ($n$ = 15, $\chi2(2)$ = 9.62, $p < .01$). The mean for the platform quality scale was 15.67 (SD = 4.99) and the mean for the content quality scale was 16.57 ($SD$ = 5.29). These scores did not vary significantly between men and women ($t(64)$ = .20, $p$ = .84; t(63) = .46, p = .64), by main occupation (F(2) = .09, $p$ = .92; F(2) = .18, $p$ = .83) or type of institution in which participants worked (F(3) = .69, $p$ = .56; F(3) = .53, $p$ = .67). However, the mean score for the platform quality scale differed significantly between participants below 45 years old ($M$ = 16.61, $SD$ = 4.67) and participants who were 45 or older ($M$ = 13.50, $SD$ = 5.73, $t(64)$ = 2.41, p < .05). For the content quality, the mean did not differ significantly by age ($t(63)$ = 1.88, $p$ = .06). The mean score for the platform and content quality scales also differed significantly by nationality (F(65) = 5.10, $p < .01$ and F(64) = 5.20, $p < .01$), with significant differences between Polish ($M$ = 11.59, SD = 5.98 and $M$ = 12.12, $SD$ = 6.91) and Spanish participants ($M$ = 17.14, $SD$ = 4.15 and $M$ = 17.94, $SD$ = 4.06) and a mean difference of 5.56 (95%CI = 1.84–9.27) and 5.82 (95%CI = 1.90–9.75). The difference in content quality was also significant between Irish ($M$ = 18.71, $SD$ = 1.50) and Polish participants ($M$ = 12.12, $SD$ = 6.91), with a mean difference of 6.60 (95%CI = 0.66–12.53). An analysis of Polish participants open-ended responses to the question "Any other comments you want to add? Please specify as much as possible" showed that most of them mentioned technical problems (such as the impossibility of watching a linked video or the fact that some links took a long time to load). One person highlighted the fact that the available resources were in English and not in Polish, which could have affected their view of the course.

A total of 74 fragments from 66 participants were extracted. Table 4 shows the system of category distribution, including a verbatim example. S1 Table included in Supplemental material shows the contingency table for the distribution of these categories across sociodemographic variables.

## Study III

Fig 2 shows the results for the NA. S2 Table provided as supplementary material shows the categories for each of the actions linked through NA. As can be seen, "breaking the silence", "excessive compliance", "familiar dynamics", "actions to promote disclosure" and "school as a

**Table 3. Emerging categories to describe victims' school experiences.**

| Category | Definition | Verbatim example[a] | n (%) |
|---|---|---|---|
| Empathy | Ability to share feelings with children, allowing them to show attention, concern, help and trust. | *"That no one came to see why I wasn't playing or why I was sad. In the end, the least important thing is why, if no one cares that you are sad, they will hardly be able to support you with a sexual abuse problem of any kind."* (I1) | 4(4.93) |
| Observation | Look at children and/or their environments closely to identify behaviour or changes in these to investigate the reason, considering that a possible cause could be victimization. | (. . .) *"The teacher could have noticed, (. . .) I used to do drawings of birds that committed suicide (. . .) at school I didn't want to go to the toilet."* (I5) | 12 (14.81) |
| Adultcentrism | Children's perception of the superiority of the adult that forces them to be subordinate and transmits to them that their voice and opinions are not as important as those of adults. | *"We were kids, we were in another category."* (I1) | 4(4.93) |
| Giving value to emotions | Space and importance given to the expression and management of feelings. | *"Let's just say it was very functional, let's just say it was like it didn't matter and the issue was to move on. There was therefore a total silence about personal feelings or pains."* (I1) | 11 (13.58) |
| Breaking the silence | Actions that promote talking about sexual victimization, going against the taboo that surrounds the subject. Accept and help victims without pathologizing them. | *"First of all, be aware that this is happening. If it's not you. Then you have to look without judgment."* (I4) | 8(9.88) |
| Actions to promote disclosure | Items that could help a victim of sexual abuse ask for help or state what is happening to them. | *"I remember that moment when I could almost have been helped. It helped me that point in which the teacher asked me what is wrong with you, but I was lacking, then, intimacy and perhaps the teacher should have insisted a little more. Insisted, and maybe asked the right question."* (I2) | 3(3.70) |
| Institutional work | Union, communication and teamwork at teaching level, so that the entire institution agrees on how to address sexual victimization. | *"The school could have done a lot if it worked as a team. That is, if it worked with the knowledge of the environment in which I lived. Having data, knowing, knowing, observing. . ."*(I3) | 6(7.41) |
| School as a refuge | School understood as a place of safety, freedom or refuge for children and adolescents who experience victimization outside of this context. | *"My liberation was to get to school. My liberation was when, even at 14 years old, they gave me an award out of 500 children for the best poem about a tree. My pride was to see that poem on the front door of the school, that one I had written hanging there. Of course, it transforms you."* (I3) | 8(9.88) |
| Isolation from peers | Feeling different, distant from their peers. | *"I remember that the relationship with colleagues also changed. I changed, I was no longer a child like the others and then the whole relationship with the classmates began to worsen (. . .) I was no longer integrated, they also treated me differently. (. . .) because I was different, because I was sad, because I was not interested in studying, but at the same time I was not one of the bad guys who was not interested in studying because they wanted to party. (. . .) there is something that is palpable that is different in you and that is rejected."* (I2) | 4(4.93) |
| Family dynamics | Behaviour of family members and bond with the child. | *"Well with a new country, with a sentimental relationship of marriage between my mother and her brother that we could not mention, that every month was full of conflicts because this situation was unsustainable and every month they took into account my grandma when she found out."* (I4) | 8(9.88) |
| Excessive compliance | Submissive attitudes and wanting to please or be reinforced. | *"How could they have helped me if I was the perfect girl, the good student, the one who won the poetry awards, the one who knew how to recite, the one who, as they could have seen if I behaved badly, was not well."* (I3) | 7(8.64) |
| Work with perpetrators | If there are victims, it is because there are perpetrators and it is also necessary to focus on them. | *"If you want to do something of this type, in person or with someone, you have to find a lot of training for abusers. (. . .) I don't think so, I'm sure. With statistics, we always talk about the victims."* (I2) | 6(7.41) |

[a] I = Interviewer.

refuge" were least connected with the actions proposed by school staff, showing only one or two links. Out of the needs expressed by victims, "observation", "whole school approach", "empathy", "emotions" and "adultcentrism" were related with three or more actions described

**Table 4. Categorizing system for the actions described by school staff.**

| Category | Definition | Verbatim example | *n*(%) |
|---|---|---|---|
| Reporting | Actions related to activating protocols or procedures to communicate a suspicion or disclosure | *Now I know that suspecting a case of child abuse is enough to report it, and if I find myself in this situation I would know how to act (. . .)* | 5(6.76) |
| Detection | Actions related to signs and indicators to identify children who are potentially at risk | *A more in-depth look at the symptoms of children's behaviour, greater attention to peer relationships (. . .)* | 14 (18.92) |
| Everyday actions | Actions that can be included in the regular school dynamics | *An introduction of the calming space for kids to calm themselves* | 21 (28.38) |
| At school level | Actions that involve several stakeholders within schools | *(. . .) I will inform management about all the tools and resources that the course has provided* | 11 (14.86) |
| More than one | Actions that could be assigned to more than one of the previously described categories | *Greater awareness of the activities undertaken by the school* | 16 (21.62) |

by school workers. Most actions that respond to victims' needs were everyday actions. As expected, "detection actions" were related with "observation". Giving value to children's disclosure and placing them in a similar level than the adult was linked with actions classified as "report".

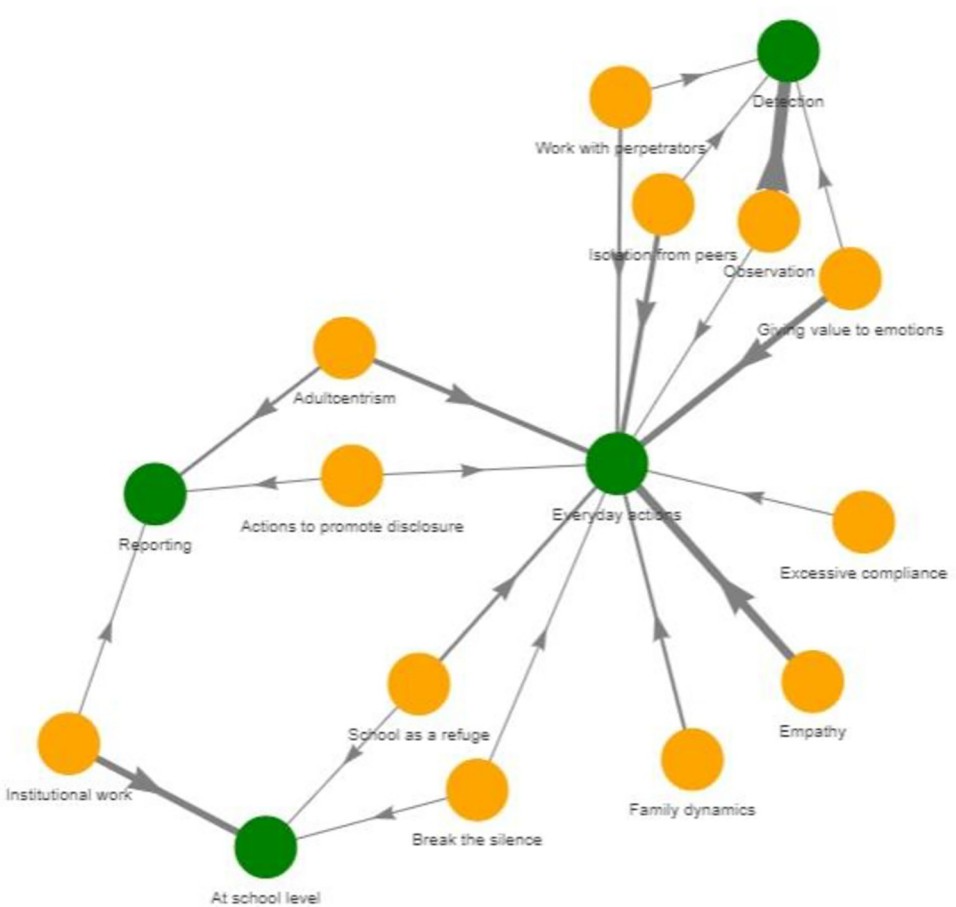

**Fig 2. Network analysis linking actions proposed by school staff (yellow nodes) with emerging categories of child sexual abuse victims (green nodes).** Edges' widths are proportional to the number of links. Interactive alternative versions of this plot are provided as supplemental material in https://osf.io/7q3rt/.

## Discussion

The present study analyses the views of education staff and woman victims of child sexual abuse on an online course for the prevention of violence against children. The objectives set at the beginning of the investigation have been achieved across the three studies conducted under the research.

Study I presents the victims' views on the role that the school had for them during the abuse and describes the different elements that impacted in their well-being at that time, highlighting those that would have helped them. This evidence respond to the first aim of the current research, i.e., to explore how victims of sexual abuse during childhood remember the school's role in their lives at the time, to try to find out what helped them and what made it harder for them. The analysis of the victims' interviews shows a lot of emphasis on the need for deeper observation of children's behavior to detect abuse. This appears especially important in sexual abuse, since this type of victimization generally does not leave any sign or mark on children's bodies [76], particularly when the abuser is a member of the family [77]. Some victims might show a lower academic performance or higher tendency to school absenteeism [12]. Hence, adults working in the educational context should consider this type of behavior a potential warning sign of abuse, rather than common "adolescent rebel behavior", as one of the participants of our study declared. However, it is crucial to take into account that other children might concentrate more on their studies while suffering abuse as a way of coping with the traumatic experience [78]. This is also something brought up by one participant of in Study I, who found in the school a safe place in contrast to what she was living at home. The fact that different victims may cope differently with the abusive experience strengths the need for a real bond between school staff and children, as an effective way to spot potential victims and provide them with help.

The value of emotions was the second category that was most frequently mentioned by victims during the interviews, principally referring to women's perception that school staff members did not value their feelings. It should be considered that during the school years of the interviewed women (in the 1970s and the 1980s) elementary education in Europe, and especially in Spain, tended to focus on cognitive competences and did not address the social and emotional skills of the children or the teachers [79]. So, it is very likely that the school members were not trained nor invested in taking care and paying attention to the pupils' emotional needs. In the current context, this finding seems consistent with proposals of schools caring for students' emotions and giving them the opportunity to share their concerns on a regular basis [32,33]. Other needs expressed by the victims during the interviews were previously identified in other investigations, such as the importance of a private, safe space to share their feelings in the school and establishing a bond of confidence with education staff [24]. All of these attitudes may translate in opportunities to tell, making the disclosure less rare. Other studies have reported similar consequences to those expressed by the interviewees, such as poor relationships with peers during their school years or behaving in a submissive way [80], which can serve as indicators or warning signs of sexual abuse [81]. Victims also expressed difficulties with their peers in school [82]. Since some of the victims feel that what is happening is wrong, they might distance themselves from their peers in order to hide the abuse and protect themselves from the judgement of their classmates.

Regarding the specific proposals, the interviewees repeatedly mentioned the importance of schools in the disclosure of abuse, highlighting their role in the implementation of new strategies for victims to break the silence. Some of these suggestions included: a) talking openly about child sexual abuse, educating on what it is and what children can do if they are victims; b) promoting relations between children and trustworthy adults outside their family circle so

that they have somebody to turn to if the abuse is being perpetrated by a family member; and c) avoiding labelling and stigmatizing victims of child sexual abuse, since this can hamper disclosure because of the shame and the taboo. In addition, the interviewed women also proposed working with aggressors. This is an especially important suggestion considering that child abusers tend to be close to the victims [83], but training sessions and workshops do not usually target them.

In relation to the second objective, Study II presents the assessment of the training at European level by educational staff and sexual abuse victims. The latter assessed specifically the module on child sexual victimization. Regarding the general feedback of the course, school staff members who were over 45 years old gave a lower score to the platform than younger participants, even though the content was highly valued by all users regardless of their age group. This might be explained by the fact that older participants may have been less familiar with the use of online platforms for education purposes and thus feel more uncomfortable with the environment [84]. Some studies reported teachers tended to prefer direct colleague or external professional conversations than information found online [85]. It could also be due to the fact that this is an introductory course and older participants might already have been trained in these topics. Therefore, the information may appear less valuable and necessary to them.

There was some variation between the answers of participants from different countries. Spanish and Irish school staff gave a higher score to the training course than Polish participants. The cultural differences might have influenced their perception of the course, and the content might be better adapted to some contexts than others. However, a Polish expert worked across the entire process of course development and their views and opinions were included to represent all the nationalities involved in the course creation. The complementary analyses of open-ended questions of Polish (see the Results section) participants also shed light on these findings, showing that some technical problems may have influenced this perception.

Concerning the objective 2a, the impact of the course in the prevention, detection and reporting of child victimization was assessed throughout the strategies that school staff members stated they would implement after taking the course to better protect the children they worked with. The most frequently mentioned changes were closer observation of children's behavior and everyday actions to prevent violence against children (like introducing a calming space within the classroom or including some specific content in sexual education workshops). Both changes were expressed through actions related to giving importance and more value to children's perceptions, feelings and experiences. This is directly connected to what interviewed victims reported that they had not received from their teachers during their school years (as shown by the findings in Study III).

Teachers also expressed their intention to apply concrete strategies to deal with disclosures and suspicions, particularly in reporting. This could also be an indicator of the positive impact of the training, as some authors previously found that specific training on violence prevention increased early reporting of possible cases of victimization among school staff [54,86].

The objective 2b (i.e., to test to what extent the actions proposed by school staff members meet the needs described by victims regarding the school's role when they suffered child sexual abuse) was achieved through Study III. The study shows that there were many connections between the desires expressed by the victims and what staff who took the course wished to implement after it, such as the previously mentioned objective of giving more importance to children emotions and paying more attention to children's actions. Other important connections relate to changes linked to the school staff's own views on children's rights, the importance given to their feelings and perceptions, the need to believe them when they disclose and report without trying to prove whether they are telling the truth. These reactions are effective for the victim's healing process [47] and may also encourage other victims to ask for help [87].

Topics that are closer to the victim and their environment appear to be less frequently mentioned by school staff, such as the child being extremely compliant and paying more attention to family dynamics [88]. In this line, "actions to promote disclosure" and "breaking the silence" were seldom mentioned by school staff, which could be explained by the fact that many of the participants asked for more tools to implement activities in the classroom after the course and might have felt unprepared to talk openly and directly about bullying and abuse.

In sum, the main contribution of this research is the evidence of the positive impact that offering teachers specific training for the prevention of violence can have in society. The study promotes a way of assessing training and intervention including, not only the perspective of the expert and the target audience, but also the victims and other stakeholders' views. Specifically, it encourages the inclusion of the victims' perspective in research on violence, since the experience can be crucial for preventing children's victimization and yet victims are rarely interviewed for this purpose.

The victims' views are also a big contribution to the child sexual abuse prevention field, since many of their opinions do not only relate to the course but directly address situations that can be found in European classrooms nowadays. In this sense, their opinions can also be taken as advice and can point the direction in which prevention should develop in the future.

Study III also represents an example of a way in which NA can be used to combine qualitative and quantitative indicators to evaluate education programs or other type of interventions and obtain meaningful insights. Finally, this investigation opens the door to future lines of research related to the evaluation of prevention training from a realistic and practical point of view. Including the points of view of victims and finding new ways to test educators' trainings can be crucial for the effective prevention of violence against children from school.

## Future research

The current research composed of three studies is an innovative proposal because it presents a free online intervention aimed at reaching any school staff member in Europe interested in developing skills and strategies to comfort and protect children from violence. However, it is also ground-breaking because assessment of the training course included the perspective of experts, school staff and victims in an integrative approach. The methods used to test the intervention combine some standardized quantitative indicators with the depth provided by qualitative techniques. Finally, the study combined the views of each type of participants through a novel method that provided meaningful, understandable insights.

Nevertheless, several limitations should be pointed out. First, it is important to consider the challenge of generalizing qualitative findings, as they tend to depend largely on the context in which a phenomenon takes place [89]. We expect that a detailed description of the situation in which this study took place would encourage future research to assess to what extent the findings are likely to be replicated in another context. Second, we used a convenience sample, so it is possible that only highly motivated people agreed to participate. Future research could test the effect of this type of training in a population including school staff and victims with different levels of motivation. Likewise, it should be taken into account as a limitation that what the adult informants tell about their past experiences could have changed, from the moment of their experience to the moment of this investigation. In this sense, we consider that if the informants had been children at the time of the investigation, different results could have been found. So, future investigations may recruit children who have suffered sexual abuse and live in safe conditions during the research, only if the ethical aspects that could challenge such a project can be warranted. Another important point to consider is that the different training

and preparation teachers and school staff must receive in each country may explain some of the findings, and even differences between universities in one region may have influenced the answers of participants. Finally, only victims of sexual abuse participated in this study, they were all women and only from Spain. Although this could provide important information on specific aspects that could contribute to making this type of victimization less hidden and taboo, future research should include victims of other types of violence, men and other genders, and from different countries to check whether the actions proposed by school staff members also meet their needs.

We hope that our project motivates other research teams to assess their intervention with strategies that include the voices of every type of stakeholder involved and enable the synthesis of quantitative and qualitative information, such as NA.

## Supporting information

**S1 Fig. Distribution of categories across interviews.** As expected, some categories are more frequent in individual interviews (e.g., category 4 in interview 2), since each individual was most sensitive about particular issues or proposals. However, most categories are present in all interviews, showing that the system tries to balance the presence of common concepts across interviews (similarity) and respects the particularity of each discourse (diversity). (DOCX)

**S1 Table. Contingency table for study II categories and sociodemographic variables.** (DOCX)

**S2 Table. Links between findings of study I and study II.** (DOCX)

**S1 Striking Image.** (HTML)

## Acknowledgments

We would like to warmly thank the people who participated in the three studies and those who implemented the knowledge extracted from the SAVE Online Training Programme. We also would like to thank our partners, represented by Moya O'Brien (ICEP Europe, Ireland), Dr. Gianluca Gini (Università degli Studi di Padova, Italy) and Renata Szredzińska (Fundacja Dajemy Dzieciom Siłę, Poland) for their collaboration in the development of the SAVE Project.

## Author Contributions

**Conceptualization:** Ana M. Greco, Noemí Pereda.

**Data curation:** Ana M. Greco, Carla González-Pío, Marina Bartolomé.

**Formal analysis:** Ana M. Greco.

**Investigation:** Ana M. Greco, Carla González-Pío, Marina Bartolomé.

**Methodology:** Ana M. Greco.

**Project administration:** Ana M. Greco, Carla González-Pío.

**Resources:** Ana M. Greco, Carla González-Pío, Noemí Pereda.

**Software:** Ana M. Greco.

**Supervision:** Noemí Pereda.

**Validation:** Noemí Pereda.

**Visualization:** Ana M. Greco.

**Writing – original draft:** Ana M. Greco, Carla González-Pío, Marina Bartolomé.

**Writing – review & editing:** Ana M. Greco, Carla González-Pío, Marina Bartolomé, Noemí Pereda.

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
