## [Decision Letter · Decision Letter 0]

26 Apr 2022

PONE-D-21-22475

How can school help victims of violence? Evaluation of online training for European schools’ staff from a multidisciplinary approach

PLOS ONE

Dear Dr. Greco,

Thank you for submitting your manuscript to PLOS ONE. After careful consideration, we feel that it has merit but does not fully meet PLOS ONE’s publication criteria as it currently stands. Therefore, we invite you to submit a revised version of the manuscript that addresses the points raised during the review process.

The manuscript has been evaluated by three reviewers, and their comments are available below.

As you will see, the reviewers are positive about the work and have raised a number of concerns that need attention, and they request additional methodological detail, analyses, and discussion of the data.

Could you please revise the manuscript to carefully address the concerns raised?

We look forward to receiving your revised manuscript.

Kind regards,

Vanessa Carels

Staff Editor

PLOS ONE

Journal Requirements:

2. Please include additional information regarding the interview guide used in the study and ensure that you have provided sufficient details that others could replicate the analyses. For instance, if you developed an interview guide as part of this study and it is not under a copyright more restrictive than CC-BY, please include a copy, in both the original language and English, as Supporting Information.

3. Peer review at PLOS ONE is not double-blinded (https://journals.plos.org/plosone/s/editorial-and-peer-review-process). For this reason, authors should include in the revised manuscript all the information removed for blind review.

4. Please change "female” or "male" to "woman” or "man" as appropriate, when used as a noun (see for instance https://apastyle.apa.org/style-grammar-guidelines/bias-free-language/gender).

5. Please provide additional details regarding participant consent. In the ethics statement in the Methods and online submission information, please ensure that you have specified whether consent was informed.

6. Thank you for stating the following financial disclosure: "No"

7. Thank you for stating the following in your Competing Interests section: "No"

8. In your Data Availability statement, you have not specified where the minimal data set underlying the results described in your manuscript can be found. PLOS defines a study's minimal data set as the underlying data used to reach the conclusions drawn in the manuscript and any additional data required to replicate the reported study findings in their entirety. All PLOS journals require that the minimal data set be made fully available. For more information about our data policy, please see http://journals.plos.org/plosone/s/data-availability.

9. We note that you have stated that you will provide repository information for your data at acceptance. Should your manuscript be accepted for publication, we will hold it until you provide the relevant accession numbers or DOIs necessary to access your data. If you wish to make changes to your Data Availability statement, please describe these changes in your cover letter and we will update your Data Availability statement to reflect the information you provide.

10. Please upload a new copy of Figure 1 as the detail is not clear. Please follow the link for more information: https://blogs.plos.org/plos/2019/06/looking-good-tips-for-creating-your-plos-figures-graphics/" https://blogs.plos.org/plos/2019/06/looking-good-tips-for-creating-your-plos-figures-graphics/

11. We note you have included a table to which you do not refer in the text of your manuscript. Please ensure that you refer to Table 2 in your text; if accepted, production will need this reference to link the reader to the Table.

12. Please include your tables as part of your main manuscript and remove the individual files. Please note that supplementary tables (should remain/ be uploaded) as separate "supporting information" files

13. Please upload a copy of Supporting Information Figure A1 which you refer to in your text on page 11.

Reviewers' comments:

Reviewer's Responses to Questions

**Comments to the Author**

1. Is the manuscript technically sound, and do the data support the conclusions?

Reviewer #1: Yes

Reviewer #2: Yes

Reviewer #3: Yes

2. Has the statistical analysis been performed appropriately and rigorously? 

Reviewer #1: Yes

Reviewer #2: Yes

Reviewer #3: I Don't Know

3. Have the authors made all data underlying the findings in their manuscript fully available?

Reviewer #1: No

Reviewer #2: Yes

Reviewer #3: Yes

4. Is the manuscript presented in an intelligible fashion and written in standard English?

Reviewer #1: Yes

Reviewer #2: Yes

Reviewer #3: Yes

5. Review Comments to the Author

Reviewer #1: The article approaches an extremely relevant topic for the field of Health and Education. Although the intervention is introductory, it is essential training for education professionals, who often feel helpless and with limited knowledge on the subject. Another strong point of the intervention is the performance of a pilot study, which allows the researcher to make the necessary adaptations for the aim study.

For each part of the article, there is a complete description of all the elements required for an empirical study, including ethical recommendations. Data analysis is consistent and appropriate to the study, and the combination of qualitative and quantitative data is adequate for this type of investigation, which addresses the issue of sexual violence, a complex phenomenon.

Below I present my suggestions:

Instruments:

In the Study I, the authors mention that the participants were also asked about several characteristics of the course content and they were invited throughout the interview to make comments or suggestions. I suggest that the authors inform at what point in the research these participants had access to the course content. I also suggest they make it clearer that this study analyses one module of the intervention related to sexual abuse (before the presentation of the goals).

Discussion:

In the first paragraph of the discussion, the authors wrote that the “the present study analyses the views of education staff and female victims of child sexual abuse on the role of the family and school environment in child protection”, different from abstract and introdution. I suggest that the authors standardize the aim of the study.

On line 308, I suggest that the authors inform that the discussion refers to Study I as they wrote at the beginning of lines 337 (Study II), and 366 (Study III).

In the data analysis, the authors mentioned that they extracted 74 fragments (I couldn't check because table 4 is not available for my viewing). However, few results were discussed, especially about the impact of the course (only one paragraph). I suggest that the authors explore this data further in a more in-depth analysis.

Figure and Tables:

- I didn’t have access to the Figure A1, which was provided as supplementary material.

- I didn’t have access to the Tables 2, 3, 4, and 5. Also it is important to mention where the tables 2, 3, and 4 need to be included in the text.

In view of this, I recommend that the authors make the necessary adjustments for the publication of the text.

Reviewer #2: Thank you for an interesting article. The idea of your article is of great importance knowing the number of digital courses and online resources that are developed to support teachers in their encounters with children experiencing violence without being evaluated. We therefore do not know much about these programs’ effectiveness, availability for teachers, and to what extend they meet the teachers’ and the victims’ needs. Your design is creative and well explained. The article would benefit from a theoretical framework to provide a deeper analysis of the data and a more thorough discussion. For the meantime, the discussion part is more or less descriptive than discussive.

Reviewer #3: This interesting work focuses on a topic of interest in the areas of health, education and psychology, and has the following objectives: to analyze how victims of sexual abuse during childhood remember the school's role in their lives at the time, to try to find out what helped them and what made it harder for them; to assess the potential of an online training course (SAVE) developed at European level, through the feedback given by school staff members and victims of child sexual abuse; to assess schools’ potential in the prevention, detection and reporting of child victimization through changes that the staff of educational centers would include in their everyday practice as a result of the course; and to test to what extent the actions proposed by school staff members meet the needs described by victims regarding the school's role when they suffered child sexual abuse. The online course used and called SAVE was created by a partnership of five European entities, which gives an important packaging to the contribution of this research.

In general, the article is well written and articulated. The review of the scientific literature carried out by the authors to support the work seems correct and pertinent to me, and at the same time I consider that the theoretical introduction is of quality, although I recommend that previous teacher training and intervention programs be reviewed in greater depth in matters of maltreatement and abuse, because there are very interesting contributions both nationally and internationally that are not named in the article.

The description of the methodological part is complete regarding the procedure. The authors could add more data about the samples and the instrument used.

The results are well structured and interesting. It is very commendable that authors have been able to articulate three studies to give a joint response to the objectives of interest. However, and unfortunally, I could not accurately evaluate this section since in the information offered to me are missed the results Tables. The figure that is presented must be improved so that its visualization is more comprehensive and letters and figures do not overlap.

The discussion is also well articulated. The authors could plan to formulate clear hypotheses and organize the discussion around them. Future lines of research are appreciated. It is recommended that the concrete contribution and practical application of the work be highlighted.

6. PLOS authors have the option to publish the peer review history of their article (what does this mean?). If published, this will include your full peer review and any attached files.

Reviewer #1: **Yes: **Grazielli Fernandes

Reviewer #2: No

Reviewer #3: No

---

## [Author Response · Author response to Decision Letter 0]

10 Jun 2022

Editor's comments:

Thank you very much for your feedback about our manuscript “How can school help victims of violence? Evaluation of online training for European schools’ staff from a multidisciplinary approach”. We also would like to thank the reviewers for their positive feedback and their suggestions. We have now revised the work according to your comments and those from the reviewers, which have significantly improved the quality of the manuscript in our opinion.

We are pleased to resubmit the revised version of the paper, along with the letter responding to each point raised by the reviewers (labeled as 'Response to Reviewers'). As you asked in your email, we are submitting a marked-up copy of the manuscript highlighting changes made to the original version with 'Track Changes' function, labeled as 'Manuscript_revised’. We are also attaching and an unmarked version of the revised paper without tracked changes, labeled as 'Manuscript'.

 Hereby we detailed the missing information, as asked in your e-mail:

• Financial disclosure: This study was funded by the European Commission through their Erasmus+ program, strategic partnership K201, 2018 call (Grant number: 2018-1-ES01-KA201-050287). The funders had no role in study design, data collection and analysis, decision to publish, or preparation of the manuscript. Author Carla González-Pío received a salary from the funder.

• Data Availability Statement: The database and code to reproduce the current study are available in https://osf.io/7q3rt/. Due to the sensibility of data, personal information has been removed. All this information has been included in page 14 of the manuscript.

• Competing interests: The authors have declared that no competing interests exist.

In addition, we would like to thank you very much for reminding us about all the journal requirements. In order to comply with them, we have now:

1. Adapted the format of our manuscript to the guidelines provided

2. Added information about the interview guide used in the study and made it available

3. Included the information previously removed for blind review

4. Made the suggested word changes

5. Specified information about the consent provided by participants

6. Included the tables in the text and provided the missing supplementary material 

We hope the reviewers and editors will be satisfied with the revision of the original manuscript.

Kind regards from the research group.

Reviewers' comments:

Reviewer #1:

Thank you very much for highlighting the relevance of our work, for your feedback and appreciations. Thank you very much as well for the specific comments in the table attached. We included all changes suggested and highlighted modifications in the revised version of the manuscript. Specifically: 

- Line 42, the sentence was replaced by “As far as children and adolescents are concerned, the developmental victimology theory suggests that children and adolescents are vulnerable to the same forms of victimization as adults (like robbery or assault) and other forms that are specific to their age group related to the dependency from the adult (such as, physical abuse or neglect by their caregivers)”. 

- Line 91, we included the reference to the mentioned dissertation through the following statement “It has been shown that sexually abused children who feel safe and communicate effectively with their parents and their protection network tend to believe that adults and the world around them are trustworthy, and therefore may be more prone to disclose their experience of abuse.”

- Line 120, we added further details about the content included in the resilience module.

- Line 122: we specified the additional materials available at the end of SAVE course.

- Line 530: We included a statement under the limitation section highlighting what you pointed out about recruiting directly children or considering that victims experience may had change over time. The text reads “it should be taken into account as a limitation that what the adult informants tell about their past experiences could have changed, from the moment of their experience to the moment of this investigation. In this sense, we consider that if the informants had been children at the time of the investigation, different results could have been found. So, future investigations may recruit children who have suffered sexual abuse and live in safe conditions during the research, only if the ethical aspects that could challenge such a project can be warranted.”

- We would like to clarify that there were “five” European entities members of SAVE (two partners were from Spain), so even if there were four participating countries, there were five institutions. We clarified this in the text, thanks to your comment. 

- We modified the paragraph starting in line 165 to explain how we dealt with differences in teacher’s training across countries. We also included a statement under the limitation section for readers to consider this aspect when interpreting the results, which reads “Another important point to consider is that the different training and preparation teachers and school staff must receive in each country may explain some of the findings, and even differences between universities in one region may have influenced the answers of participants.”

- We moved the reference about conventional content analysis principles to make it clear in which approach we were based to perform our qualitative analysis.

- In the limitation section we also mentioned the fact that all participants of Study I were women and from Spain (line 538)

- We mentioned the suggested reference (Ellis, 2018), which in fact guided part of our work in line 166.

- We included the level of studies achieved by each of the participants in Study I.

- In line 310 we detailed all R packages used for the analyses (“In addition to the standard packages, “psych” (Revelle, 2022) and “fmsb” (Nakazawa, 2022) was used to assess psychometric properties, “ggplot2” (Wickham, 2016) to plot analyses. “network” (Butts, 2008; 2015) and “igraph” (Csardi & Nepusz, 2006) to perform and plot network analyses. R Software [60] was used to perform all analyses). We also made the code and dataset available so the study can be reproduced here: https://osf.io/7q3rt/ (and stated it in line 314).

- We modified Figure 2 as you and the editor suggested 

- We added a paragraph in the discussion about the fact that not all victims perform poorly in school, including the reference you proposed. It states: “Other studies also showed that children might concentrate more on their studies while suffering abuse as a way of coping with the traumatic experience (Selvik, 2020). This is also something brought up by one participant of in Study I, who found in the school a safe place in contrast to what she was living at home. The fact that different victims may cope differently with the abusive experience strengths the need for a real bond between school staff and children, as an effective way to spot potential victims and provide them with help.”

- We included a comment about the results of the very recent reference you suggested (Selvik & Helleve, 2022). 

- We moved the sentence “An analysis of Polish participants open-ended responses to the question “Any other comments you want to add? Please specify as much as possible” showed that most of them mentioned technical problems (such as the impossibility of watching a linked video or the fact that some links took a long time to load). One person highlighted the fact that the available resources were in English and not in Polish, which could have affected their view of the course” to the results section, as suggested.

Thank you very much for your suggestions regarding the Instrument. We modified the Procedure section to clarify when participants were presented with the content of the course. We also included the specific aim of assessing the sexual victimization module through the opinion of victims of sexual abuse. 

Thank you very much for your feedbacka about the discussion. We modified the phrase (line 401) and we restructured the discussion section linking each study results with the aims proposed. We reformulated the main objective in the discussion and added “Study I” in the text, which makes the article way more clear and easier to read.

We have now correctly included the tables within the text where they should be placed. We agree that currently the text corresponds better to the data. We decided not to go further on the analyses as we focused only in our main findings. We also updated the supplementary material and included all supporting information in https://osf.io/7q3rt/

Thank you once more for helping us to improve our work.

Reviewer #2:

Thank you very much for your suggestion. We have framed our work within the basis of the theoretical framework developed by Prof. Finkelhor's, i.e., Developmental victimology. We have also modified the discussion and introduction including new references pointed out by other reviewers, which we agree that enriched the framework and the arguments. 

Reviewer #3:

Thank you very much for your careful appreciation of our work. Following your suggestions, we included more studies about teacher trainings and intervention programs (e.g., Gün et al., 2022). We also referred to a meta-analytic review about the components of school programs that successfully tackle child abuse (Gubbels et al., 2021). We also added further details about the sample of study I (age range, level of studies achieved) and study II (level of studies achieved, professional profile). We also made all instrument protocols available and specified some details (administration platform, item wording) in the instrument section.

We also modified the figure and included the tables in the same manuscript document. We are sorry that you were not able to review this material and hope you find it relevant.

Finally, we included more specific information about the contributions of this research and propose new lines of research for the future. Regarding the organization of the discussion, we agree it could be better presented. Instead of making new hypotheses, we decided to organize the discussion around the objectives described at the beginning of the investigation. We believe the text is now clearer and more coherent. 

Thank you again for helping us to improve the manuscript.

---

## [Editor Report · Decision Letter 1]

28 Jul 2022

How can school help victims of violence? Evaluation of online training for European schools’ staff from a multidisciplinary approach

PONE-D-21-22475R1

Dear Dr. Greco,

We’re pleased to inform you that your manuscript has been judged scientifically suitable for publication and will be formally accepted for publication once it meets all outstanding technical requirements.

Kind regards,

Dylan A Mordaunt, MD, MPH, FRACP

Academic Editor

PLOS ONE

Additional Editor Comments (optional):

Thank you for your resubmission. This was reassigned to me as academic editor, after involving a different academic editor or process, initially. On the first round, three reviews were received and feedback given that major revisions were required.

The comments made by the authors have been adequately dealt with and this now meets the criteria for publication. With specific reference to the criteria for publication:

1. The report presents the results of original research.

2. Results reported have not been published elsewhere.

3. Experiments, statistics, and other analyses are performed to a high technical standard and are described in sufficient detail.

4. Conclusions are presented in an appropriate fashion and are supported by the data.

5. The article is presented in an intelligible fashion and is written in standard English.

6. The research meets all applicable standards for the ethics of experimentation and research integrity.

7. There are peripherally relevant reporting guidelines, but the authors have included multiple studies, making this an immensely rich report and I don't think further changes to structure would add value.
---

## [Editor Report · Acceptance letter]

3 Aug 2022

PONE-D-21-22475R1 

How can school help victims of violence? Evaluation of online training for European schools’ staff from a multidisciplinary approach 

Dear Dr. Greco:

I'm pleased to inform you that your manuscript has been deemed suitable for publication in PLOS ONE. Congratulations! Your manuscript is now with our production department. 

Kind regards, 

on behalf of

Associate Professor Dylan A Mordaunt 

Academic Editor

PLOS ONE